# Evaluation of Helmet Wearing Compliance: A Bionic Spidersense System-Based Method for Helmet Chinstrap Detection

**DOI:** 10.3390/biomimetics10090570

**Published:** 2025-08-27

**Authors:** Zhen Ma, He Xu, Ziyu Wang, Jielong Dou, Yi Qin, Xueyu Zhang

**Affiliations:** College of Electromechanical Engineering, Harbin Engineering University, Harbin 150009, China; mazhen@hrbeu.edu.cn (Z.M.); wangziyu@hrbeu.edu.cn (Z.W.); 1691631927@hrbeu.edu.cn (J.D.); 7042@hrbeu.edu.cn (Y.Q.); zhangxueyu@hrbeu.edu.cn (X.Z.)

**Keywords:** helmet-wearing detection, bionic spidersense system, MEMS inertial sensors, ICNN-LSTM, helmet chinstrap

## Abstract

With the rapid advancement of industrial intelligence, ensuring occupational safety has become an increasingly critical concern. Among the essential personal protective equipment (PPE), safety helmets play a vital role in preventing head injuries. There is a growing demand for real-time detection of helmet chinstrap wearing status during industrial operations. However, existing detection methods often encounter limitations such as user discomfort or potential privacy invasion. To overcome these challenges, this study proposes a non-intrusive approach for detecting the wearing state of helmet chinstraps, inspired by the mechanosensory hair arrays found on spider legs. The proposed method utilizes multiple MEMS inertial sensors to emulate the sensory functionality of spider leg hairs, thereby enabling efficient acquisition and analysis of helmet wearing states. Unlike conventional vibration-based detection techniques, posture signals reflect spatial structural characteristics; however, their integration from multiple sensors introduces increased signal complexity and background noise. To address this issue, an improved adaptive convolutional neural network (ICNN) integrated with a long short-term memory (LSTM) network is employed to classify the tightness levels of the helmet chinstrap using both single-sensor and multi-sensor data. Experimental validation was conducted based on data collected from 20 participants performing wall-climbing robot operation tasks. The results demonstrate that the proposed method achieves a high recognition accuracy of 96%. This research offers a practical, privacy-preserving, and highly effective solution for helmet-wearing status monitoring in industrial environments.

## 1. Introduction

Intelligent manufacturing, as a cornerstone of Industry 4.0, is fundamentally reshaping the ways in which humans work and live [1]. With the ongoing advancement of intelligent industrialization, safety management on industrial sites is encountering unprecedented challenges. Industrial accidents not only pose risks of severe injuries or fatalities but can also significantly disrupt production schedules and negatively affect enterprise profitability [2]. According to data from the U.S. Bureau of Labor Statistics, in 2020, head injuries accounted for nearly 6% of non-fatal occupational injuries resulting in days away from work [3]. Therefore, safety helmets—among the most widely used forms of personal protective equipment (PPE)—play an essential role in safeguarding workers’ head health on construction and industrial sites. Although the majority of workers comply with mandatory helmet-wearing regulations, monitoring the proper usage of safety helmets, particularly the correct fastening of chin straps, remains a significant challenge in occupational safety management [4,5].

Traditional safety compliance monitoring methods primarily rely on visual inspection through video surveillance [6,7], which can determine whether a worker is wearing a helmet but often fails to accurately assess the correctness of its usage—particularly in complex working environments where system accuracy and real-time performance frequently fall short of safety management standards. Moreover, image-based detection requires stable lighting conditions and consistent head and facial orientations [8,9], making it poorly suited for real-world operational scenarios. Notably, facial data constitutes personally identifiable information [10], raising significant privacy concerns. Therefore, developing an efficient and privacy-preserving method for assessing helmet compliance remains a critical challenge.

Spiders possess highly developed mechanosensory systems, with their leg-based sensory organs serving as influential models in mechanical vibration perception research [11]. Further studies have shown that the anterior legs of spiders contain diverse hair-like sensory structures that enhance stimulus perception [12,13]. Inspired by the sensing mechanisms of spider leg joints, researchers have developed ultra-sensitive mechanical crack sensors [14], capable of detecting strain and vibration signals with high mechanical sensitivity. Similarly, a tunable, highly sensitive epidermal sensor has been proposed [15], drawing inspiration from the natural ability of spider leg posture adjustments to actively modulate sensory sensitivity. These examples highlight the important role of spider leg sensory systems—particularly the hair arrays—in mechanical perception. Although current bioinspired applications based on spider sensing mainly focus on sensor development, there is limited exploration at the level of sensory strategies, indicating a need for further investigation into biomimetic sensing frameworks.

In recent years, with the advancement of sensing technologies, MEMS (Micro-Electro-Mechanical Systems) sensors have become a key area of research in industrial safety monitoring due to their low power consumption, high precision, and strong anti-interference capabilities. Particularly in dynamic detection applications [16], MEMS sensors enable real-time capture of helmet posture and movement changes, offering more accurate recognition of wearing states compared to traditional approaches. Continuous monitoring of workers’ helmet-wearing conditions can effectively detect and alert improper fastening or loosening of chinstraps, thereby mitigating safety risks associated with non-standard usage.

MEMS attitude sensors play a crucial role in various domains such as robot balance control, human motion analysis, and aircraft attitude measurement. As a subset of inertial measurement units (IMUs), these sensors are widely applied in mobile robotics [17], human–robot collaboration [18], and other fields to detect real-time movement and attitude variations [19,20,21]. MEMS attitude sensors collect data signals including angular velocity, acceleration, and magnetic field intensity, which are essential for monitoring the operational status of humans or mechanical equipment [22]. However, in long-term state detection scenarios, MEMS attitude sensors often suffer from drift and reduced accuracy, making effective data preprocessing and appropriate model selection both critical and necessary [23].

Traditional machine learning approaches have achieved some success in robot fault diagnosis but are constrained by long computation times, relatively low accuracy, and dependence on manual feature extraction [24,25]. Deep learning methods, particularly hybrid models combining convolutional neural networks (CNNs) and long short-term memory (LSTM) networks [26,27], have emerged as powerful tools for sequential signal classification due to their robust feature extraction and temporal modeling capabilities. Nevertheless, CNN architectures tend to be highly complex, involving numerous hyperparameters [28,29]. Consequently, designing adaptive CNN models is considered an optimal strategy [30,31].

This study proposes a biomimetic helmet chinstrap tension detection system inspired by the sensory hairs on spider legs. By simulating the sensing process of spider leg hair arrays, a hierarchical perception strategy is designed using multiple attitude sensors to collect helmet posture data for identifying the tightness of helmet chinstraps. The specific contributions include the following: (1) development of a helmet-wearing detection system based on a bioinspired multi-layered attitude sensor configuration modeled after spider tactile sensing, utilizing three MEMS attitude sensors to acquire workers’ head posture features; (2) design of an efficient data processing algorithm for classifying and recognizing helmet wearing states; (3) experimental validation of the system’s performance under controlled laboratory conditions. Compared to existing methods, the proposed system enables continuous and real-time detection of helmet-wearing status while preserving user privacy and overcoming limitations of conventional approaches. Figure 1 illustrates the overall research framework.

This paper is organized as follows: Section 2 introduces the bioinspired helmet chinstrap tension detection strategy based on spider leg hair arrays; Section 3 describes the implementation of the helmet chinstrap tension recognition system and presents an improved CNN-LSTM model for classification of helmet-wearing states; Section 4 details the experimental data collection procedures and testing conditions; Section 5 presents the experimental setup and results analysis; finally, Section 6 concludes this paper and outlines future research directions.

## 2. Detection Strategy for Helmet Chin Strap Tightness Inspired by Spiny Leg Bristle Sensory Mechanisms

This section investigates the feasibility of applying biomimetic design principles inspired by the sensory mechanisms of spider leg bristles to develop an effective method for detecting the tightness of helmet chin straps. The exploration focuses on analyzing the underlying principles and key constraints governing chin strap tension.

### 2.1. Biomimetic Design of the Sensory Strategy Based on Spider Leg Bristles

Spider legs are equipped with highly sensitive tactile organs capable of detecting minute vibrations and mechanical stimuli. These sensory structures exhibit exceptional responsiveness to subtle variations in force intensity. Spiders utilize such tactile signals to discern the position, size, and movement patterns of prey, thereby enabling rapid behavioral responses during predation. The tactile organs on spider legs demonstrate remarkable sensitivity, allowing for the detection of extremely small forces and vibrational cues. Each leg contains multiple sensory setae, known as trichobothria, which are structurally composed of long or short, thin hairs embedded within cup-shaped bases lined with a flexible membrane. Each individual seta provides specific information regarding the spatial orientation and motion dynamics of prey relative to the spider [32]. This heightened sensitivity is attributed to the variation in setal lengths, which confer differential responsiveness to diverse force magnitudes and vibrational frequencies. Analogously, in the context of safety helmets, the dynamic interactions between the helmet and the wearer’s head vary under different wearing conditions—particularly when the tightness of the chin strap changes. Detecting these variations can facilitate accurate assessment of the helmet’s fit status.

Inspired by this biological mechanism, we developed a sensor system that mimics the vibration-sensitive characteristics of three distinct types of bristles found on spider legs. Each sensor demonstrates differential sensitivity to the frequency and amplitude of mechanical vibrations, replicating the response behavior of spider setae to varying stimulus intensities. These sensors were mounted on the outer surface of the safety helmet shell to enable high-sensitivity monitoring of its vibrational response. As illustrated in Figure 2, Sensor 1 is directly attached to the helmet surface to emulate short bristles that capture high-frequency vibrations; Sensor 2 is connected via a shorter fiber rod, simulating medium-length bristles that respond to mid-range frequencies; Sensor 3 is installed using a longer fiber rod, mimicking the low-frequency sensitivity of long bristles. Detailed installation parameters for each sensor are provided in Table 1.

### 2.2. Principle of Chinstrap Tightness Recognition Under Safety Helmet

To simulate the dynamic interaction between the helmet and the wearer’s head, we employed a classical spring–damper model. In this modeling framework, the spring component represents the elastic contact forces between the helmet and the head, while the damper component accounts for the frictional forces and energy dissipation occurring during helmet–skin contact. By applying this model, we are able to simulate the relative motion between the helmet and the head under different levels of chinstrap tightness. The underlying theoretical formulation is described as follows:

(1)Dynamic Coupling and Relative Motion Model Between Helmet and Head

When the helmet is secured to the head via the chinstrap—forming either a rigid or semi-flexible connection—their coupled motion can be characterized using rigid body dynamics. Assuming the head’s motion trajectory as the reference frame, the helmet’s acceleration can be expressed in terms of its relative components.(1)$$a_{\mathrm{helmet}}=a_{\mathrm{head}}+a_{\mathrm{relative}}$$
where $a_{\mathrm{head}}$ is the true acceleration of the head, and $a_{\mathrm{relative}}$ is the relative acceleration between helmet and head.

By applying the Newton–Euler equations, the dynamic relationship of the helmet-head system is established as follows:(2)$$m_{\mathrm{helmet}}\cdot a_{\mathrm{relative}}=F_{\mathrm{strap}}-c\cdot v_{\mathrm{relative}}-\nabla U\left(x\right)$$
where $m_{\mathrm{helmet}}$ represents the helmet mass, $F_{\mathrm{strap}}=k\cdot F_{\mathrm{tension}}$ is the chinstrap tension, c is the system damping coefficient, and U(x) represents helmet displacement related to potential energy.

It can be observed that when the tightness *k* decreases, $a_{\mathrm{relative}}$ increases, alongside a marked rise in high-frequency attitude components.

(2)Attitude Response and Damping Characteristic Analysis

Modeling the helmet–head system as a mass–spring–damper system, its equation of motion is expressed as follows:(3)$$m\ddot{x}\left(t\right)+c\left(k\right)\dot{x}+k_{s}x=F_{ext}\left(t\right)$$
where c(k) is the effective damping coefficient correlated with chinstrap tightness, $k_{s}$ is the system stiffness, and $F_{\mathrm{ext}}\left(t\right)$ is the external excitation force.

By conducting frequency domain response analysis, the transfer function can be derived as follows:(4)$$H\left(\omega \right)=\frac{1}{\sqrt{{\left(k_{s}-m\omega_{2}\right)}^{2}+{\left(c\left(k\right)\omega \right)}^{2}}}$$

Assuming constants $k_{s},m,\omega$, and varying *k* from 0 to 1 in increments of 0.25, the corresponding transfer functions are plotted in Figure 3.

From the results, it is evident that when chinstrap tightness is insufficient (i.e., *k* decreases), the system damping diminishes, causing a reduction in the transfer function gain across the frequency spectrum, particularly within the low-frequency range. This implies that smaller *k* values weaken the system’s response to low-frequency signals.

## 3. Implementation Scheme of Helmet Chin Strap Tightness Recognition System

This section presents an analysis of MEMS sensor data acquisition for the recognition of helmet chinstrap tightness. It provides a comprehensive description of the mathematical processing techniques applied to the sensor data, as well as an overview of the proposed improved convolutional neural network–long short-term memory (ICNN-LSTM) model. The discussion focuses on the extraction of discriminative features associated with chinstrap tightness from the collected attitude data.

### 3.1. Definition and Analogy of Helmet Chin Strap Tightness

The objective of this study is to enable helmets to accurately distinguish between different levels of chinstrap tightness during wear. Traditional criteria for defining chinstrap tightness in safety helmets are often subjective and lack precise quantification, making it difficult to establish standardized measurement methods. To address this limitation, this section defines three distinct categories of chinstrap tightness based on equivalent tensile force measurements obtained when a 1.5 cm gap is created between the helmet’s chinstrap and the wearer’s chin. These defined categories are subsequently validated against real-world wearing conditions, as demonstrated in Figure 4.

Based on the defined chinstrap tightness criteria, the wearing status of a safety helmet can be categorized into three distinct classifications:(1)Improperly Fastened: The helmet is positioned on the head, but the chinstrap buckle remains unsecured and does not make contact with the wearer’s chin, resulting in a loose fit. When the chinstrap is pulled vertically downward using a force gauge to create a 1.5 cm gap between the strap and the chin, the measured tensile force is less than 1 N.(2)Loose Fit: Following manual adjustment of the chinstrap while the helmet is worn, pulling the strap vertically downward to generate a 1.5 cm gap results in a tensile force ranging from 1 N to 3 N.(3)Standard Fit: Similarly, after manual adjustment under wearing conditions, the tensile force recorded when pulling the chinstrap vertically downward to form a 1.5 cm gap exceeds 5 N.

### 3.2. Hardware Composition of the Helmet Chin Strap Tightness Sensing System

The proposed wearing-aware helmet system integrates MEMS attitude sensors into a standard safety helmet, as illustrated in Figure 5, which depicts the physical configuration and the sensor coordinate system. The MEMS attitude sensor consists of a three-axis accelerometerometer, a three-axis gyroscope, and a three-axis magnetometer, enabling real-time monitoring of the amplitude and frequency of workers’ head movements relative to the helmet. The sensor is securely mounted on the helmet and communicates with the data processing module via 5G Bluetooth technology. Identical sensor models with consistent parameters and precision levels are listed in Table 2. It should be noted that in this study, the sampling frequency was set to 100 kHz to ensure maximum sensor measurement accuracy. Moreover, all data used in this research were obtained exclusively from the inertial measurement unit (IMU) of the MEMS sensors. In the data processing pipeline, the three-axis angular acceleration measurements were directly utilized for feature extraction and computational analysis.

### 3.3. Data Acquisition from Helmet MEMS Attitude Sensors

The low-cost WT9011DCL-BT50 series attitude sensors employed in this study exhibit a residual gravitational acceleration reading of 1 g even after zero-point calibration. Furthermore, the initial orientation of the sensor may vary across different installations, leading to increased variability in the training data and consequently degrading the model’s generalization capability. To reduce the influence of variations in sensor installation direction and gravitational orientation on model training and prediction accuracy, this study scalarizes the three-axis acceleration vector obtained from the sensor. Let $a_{x}$, $a_{y}$, and $a_{z}$ represent the acceleration values along the *x*, *y*, and *z* axes, respectively. The sensor’s output acceleration vector is expressed as $a={\left[a_{x},a_{y},a_{z}\right]}^{T}$.

The magnitude of the acceleration vector is calculated and converted into a scalar value, which serves as an input feature to minimize measurement errors caused by inconsistent installation orientations. The computation process is formulated as follows:(5)$$∥a∥=\sqrt{a_{x}^{2}+a_{y}^{2}+a_{z}^{2}}$$

The input signal for the model is then represented by the following equation.(6)$$F_{\mathrm{input}}=∥a∥$$

In this study, a low-pass Butterworth filter with a cutoff frequency of 1 Hz, a filter order of 5, and a sampling frequency of 500 Hz was employed to preprocess the data. This type of filter is characterized by its maximally flat frequency response and is particularly suitable for applications requiring smooth frequency characteristics. The filter design is based on the following considerations:(7)$$H\left(s\right)=\frac{b_{0}+b_{1}s+\cdots +b_{n}s^{n}}{a_{0}+a_{1}s+\cdots +a_{n}s^{n}}$$
where H(s) represents the filter transfer function, $b_{i}$ and $a_{i}$ is the cutoff frequency, and *n* is the filter order.

After filtering, we use the MinMaxScaler to normalize the data, mapping it to the range [0,1]. The normalization formula is given by the following equation:(8)$$x^{\prime }=\frac{x-x_{\min}}{x_{\max}-x_{\min}}$$
where *x* s the original data, $x_{\min}$ and $x_{\max}$ are the minimum and maximum values of the data, and the output is the normalized data.

It should be emphasized that the normalization process is performed after the dataset has been partitioned into training (70%), validation (10%), and testing (20%) subsets. Specifically, the MinMaxScaler is fitted solely on the training set by utilizing its minimum and maximum values to determine the scaling parameters. These parameters are subsequently applied to transform both the training and testing subsets, thereby preventing any potential data leakage.

### 3.4. Feature Generation

Following the completion of data preprocessing, it is essential to further represent and extract relevant data features. Selecting an appropriate representation method is critically important for subsequent computational processing. In this study, the MEMS sensing unit operates at a sampling frequency of 100 kHz. To facilitate efficient data handling, the collected attitude data is divided into segments, each containing 64 data points, which correspond to one set of attitude measurements from the experimental platform. It should be emphasized that the selection of 64 data points was determined through a balanced consideration of time resolution and computational efficiency. This segment length effectively captures the dynamic characteristics associated with changes in chinstrap tightness while avoiding excessive computational complexity caused by overly long sequences. The chosen segmentation strategy supports the effective extraction of both time-domain and frequency-domain features, thereby providing robust data support for downstream analysis.

A sliding window approach is employed for data segmentation, with an overlapping window configuration to maintain temporal continuity. This method enables better capture of the temporal characteristics of the signal and is particularly valuable for real-time monitoring applications. Subsequently, time-domain and frequency-domain features are extracted from each data segment, yielding a total of 17 distinct feature indicators. Table 3 presents the names and categories of the extracted features. Finally, the feature values are normalized to construct standardized feature vectors for model training.

### 3.5. Helmet Chinstrap Tightness Recognition Framework Based on ICNN-LSTM Network

To utilize the signals from the MEMS attitude sensors in the helmet for chinstrap tightness recognition, this study proposes a classification process for helmet chinstrap tightness, as shown in Figure 6. The process begins with preprocessing the input attitude signals. Next, the dataset was divided into training (70%), validation (10%), and testing (20%) sets. The MinMaxScaler, fit on the training set, is then used to normalize all sets, ensuring consistency and preventing data leakage. Six-fold cross-validation was employed to assess the performance of the dual model. Subsequently, the proposed ICNN-LSTM model is trained using the training set for classification.

(1)Feature Extraction: Adaptive Convolutional Neural Network (ICNN)

In the model architecture, the improved convolutional neural network (ICNN) integrates classical one-dimensional convolutional operations with an adaptive rectified linear unit (ReLU) activation function to enhance the network’s representational capacity and classification accuracy. The architecture comprises multiple convolutional blocks, each consisting of a 1D convolutional layer, an adaptive activation layer, a max-pooling layer, and a dropout layer. This design facilitates the extraction of rich and discriminative features for subsequent processing. Within the convolutional layers, standard fixed kernel dimensions and filter counts are employed, specifically a kernel size of 3 and 64 filters, which enable the effective extraction of local features from the input data and capture short-term dependencies in sequential patterns. Following each convolutional block, an adaptive ReLU activation layer is introduced to further improve the non-linear expressive capability of the network. The mathematical formulation of this adaptive activation function is defined as follows:(9)$$p_{li}\left(t\right)=\max\left(0,z_{li}\left(t\right)\right)+\alpha_{li}\min\left(0,z_{li}\left(t\right)\right)$$

Let $p_{li}\left(t\right)$ represent the output of the *i*-th neuron in the *l*-th layer after applying the adaptive ReLU activation function, $z_{li}\left(t\right)$ denote the input feature values to the batch normalization (BN) layer, and let $\alpha_{li}$ be the adaptive parameter to be learned by the layer. The design of adaptive ReLU aims to enhance the network’s ability to handle negative values. By utilizing an adaptive negative slope, it maintains non-linearity while increasing the model’s ability to adjust feature distributions.

Moreover, to prevent feature redundancy and overfitting, each convolutional block also includes a max pooling layer and a dropout layer. The max pooling layer employs a pooling window of size 1 to reduce feature dimensions and enhance local feature selectivity, while the dropout layer discards 20% of the neurons in each convolutional block to further mitigate the risk of overfitting.

(2)Dangerous State Classification: Long Short-Term Memory (LSTM) Layer

The LSTM layer used in the convolutional and LSTM integrated neural network (ICNN-LSTM) is designed to extract temporal features and further model the temporal dependencies of local features encoded in the convolutional feature maps. The LSTM layer consists of multiple memory units, which regulate information updates at each time step through input gates, forget gates, and output gates. First, the output of the convolutional block is fed into the LSTM layer, where the LSTM units process the information at each time step sequentially to form a comprehensive temporal feature representation.

The output of each LSTM unit is produced by the combination of the internal memory unit $c_{t}$ and the output gate $o_{t}$. The update process of the memory unit $c_{t}$ is as follows:(10)$$c_{t}=f_{t}⊙c_{t-1}+i_{t}⊙c_{t}$$
where $f_{t},i_{t}$, and $c_{t}$ represent the forget gate, input gate, and candidate memory cell state, respectively. The forget gate determines which past information to retain or discard, the input gate controls the degree of current information storage, and the candidate memory cell state holds the new information temporarily. Finally, the output of the LSTM unit $h_{t}$ is generated through the output gate $o_{t}$, calculated as follows:(11)$$h_{t}=o_{t}⊙\tanh\left(c_{t}\right)$$

Through this gating mechanism, the LSTM can dynamically update its state at each time step, capturing long-range dependencies in the temporal data.

Lastly, all outputs from the LSTM layer are passed to the fully connected layer via a flattening operation, generating a final discriminative feature map $x_{\mathrm{out}}$. This feature map can further be input into a classifier composed of fully connected layers and a Softmax layer to obtain the classification probability outputs of the time-series data. Additionally, this feature map can be used for cross-domain feature extraction in transfer learning frameworks, enhancing the model’s generalization ability under varying data conditions.

The integration of the ICNN network with the LSTM network forms the ICNN-LSTM network, as shown in Figure 7.

Algorithm 1 describes the implementation process of the ICNN-LSTM classification algorithm.
**Algorithm 1:** ICNN-LSTM Model Training and Evaluation
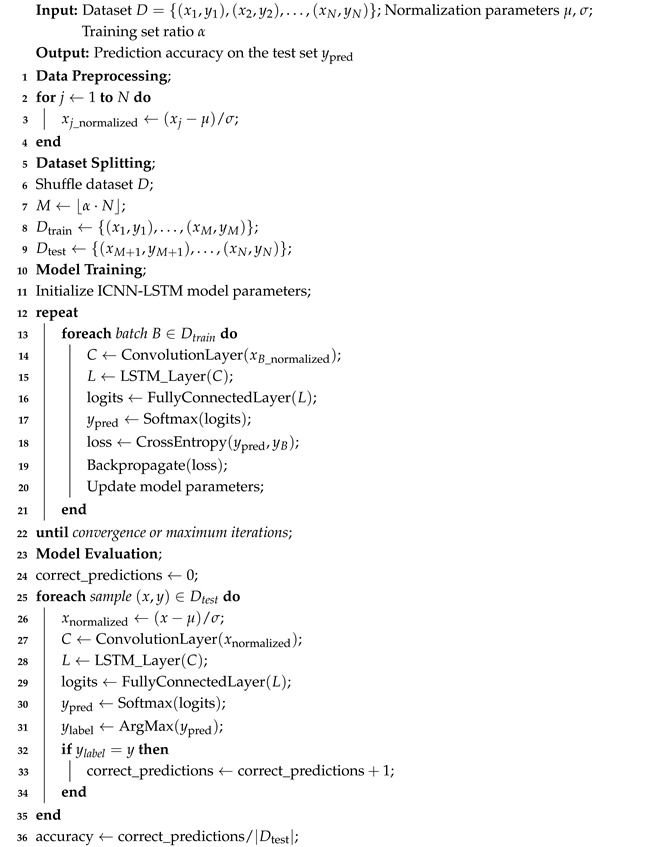


## 4. Experimental Data Preparation

### 4.1. Experimental Setup

In this section, to evaluate the effectiveness of the proposed method for sensing and classifying the chinstrap tightness state of a safety helmet, a shipboard wall-climbing robot control test system is employed to simulate real-world worker tasks. As illustrated in Figure 8, the system comprises an operator, an intelligent safety helmet, a wheeled wall-climbing robot, helmet and posture sensors, an arcuate steel plate wall, and a load module. It should be noted that the wheeled wall-climbing robot utilized in this study is based on a dual-wheel differential motion model. Permanent magnets are integrated into the inner sides of the two active wheels and one passive wheel on both sides of the robot to ensure sufficient adhesion when operating on vertical or inclined surfaces. The operator can precisely control the robot’s trajectory by adjusting the differential motion between the two active wheels.

The task simulation experiment is conducted in a controlled laboratory environment, with the steel plate and frame securely anchored. All operators are trained and demonstrate proficient and consistent performance during operation. The experimental protocol requires each operator to control the robot along a predefined path using a wireless controller. The operator must remain seated at a fixed position and maintain continuous visual monitoring of the robot and its surrounding environment until the robot reaches the designated target location, at which point the task is deemed complete.

### 4.2. Experimental Participants

The experiment was conducted in an indoor laboratory at Harbin Engineering University, where data collection was performed with a total of 10 participants. All participants were graduate students, consisting of 9 males and 1 female. Due to the male-to-female ratio in the laboratory, it was not possible to achieve gender balance in this experiment. All participants were in good physical condition on the day of the test. Before the test began, personal information such as age, gender, height, weight, and body fat percentage for each participant was recorded, as shown in Table 4.

Each participant’s experimental session was structured into three sequential phases:

Step 1: Collection of Baseline Information and Pre-training

At the beginning of the session, participants provided their baseline demographic and health-related information. Researchers then explained the purpose, procedure, equipment, and task requirements to eligible participants. All subjects underwent a pre-training session to familiarize themselves with the control joystick and practice completing the robot transportation task. Following pre-training, each participant took a 10 min rest period to ensure sustained alertness and readiness for the main experiment.

Step 2: Execution of Wall-Climbing Robot Transportation Task and Data Acquisition

Participants were required to wear a safety helmet equipped with a chinstrap secured in the standard configuration. Under controlled experimental conditions, they followed predefined operational guidelines, including remaining at the designated workstation and continuously monitoring the robot. Using the control joystick, participants guided the mobile robot along a specified trajectory to complete the wall-climbing task. The robot’s climbing speed was set at three levels—0.05 m/s, 0.1 m/s, and 0.2 m/s—with three experimental trials conducted at each speed. Throughout the task, the safety helmet continuously collected data from the integrated MEMS sensors, which were transmitted and stored on the host computer.

Step 3: Adjustment of Chinstrap Tightness and Repetition of Experiments

Participants then adjusted the chinstrap to a looser configuration and repeated the procedures outlined in Step 2, with data recorded accordingly. Subsequently, the chinstrap was fully loosened (i.e., unfastened), and the same experimental protocol was executed once more, with data saved for analysis. Each participant completed all three experimental conditions to ensure comprehensive data collection across varying chinstrap tightness levels.

## 5. Results and Discussion

### 5.1. Hyperparameter Settings

Subsequently, a CNN-LSTM network was employed to classify participants’ helmet wearing conditions. The dataset was partitioned into training (70%), validation (10%), and testing (20%) subsets to effectively evaluate model performance. Six-fold cross-validation was conducted to assess the classification capability of the dual-model architecture. The experimental results demonstrated that the proposed fine-tuning strategy, with parameters including 30 training epochs, a dropout rate of 0.2, a batch size of 32, and a learning rate of 0.001, achieved the highest classification accuracy.

(1)Learning Rate

The learning rate governs the convergence behavior of the objective function during optimization. An appropriately chosen learning rate facilitates efficient convergence to a local minimum. Based on prior empirical experience, a stepwise decreasing learning rate schedule was adopted, starting at 0.1 and reduced by a factor of 10 after each training phase. As illustrated in Figure 9, the training results indicated that after 30 iterations, with the learning rate reduced to 0.001, the classification accuracy surpassed 90%, outperforming configurations using fixed rates of 0.1 and 0.01. Moreover, both the accuracy and loss curves exhibited greater smoothness. Therefore, considering both convergence efficiency and overall accuracy, a learning rate of 0.001 was determined to be optimal.

(2)Training Batch Size

The training batch size determines how the dataset is partitioned into smaller subsets for each training iteration. If the batch size is too small, it may lead to prolonged training time and unstable convergence; conversely, if it is too large, it may result in underfitting or memory constraints. To identify the optimal batch size, we initiated the experiments with a size of 8 and progressively increased it. The experimental results, as shown in Figure 10, demonstrated that a batch size of 16 achieved the most balanced classification accuracy across all three categories. Consequently, a batch size of 16 was selected as the optimal configuration.

The other main parameters of the ICNN-LSTM model are detailed in Table 5.

### 5.2. Comparison Between Different Sensors

To demonstrate the effectiveness of the proposed ICNN-LSTM classification model, data were collected using three sensors mounted on the helmet under three different robot operation speeds. Furthermore, five state-of-the-art classification models were selected for comparative analysis, including LSTM, RNN, BP neural network, RF (Random Forest), and KNN (k-Nearest Neighbors). These five models are well established in the field, and detailed descriptions are omitted in this paper for brevity.

(1)Data Collected with Single Sensor Installation under Different Conditions

Under the single working condition scenario, sensor data from each participant were stored separately according to the sensor position—labeled as Sensor1, Sensor2, and Sensor3. The experimental tasks were conducted at three predefined robot operation speeds, and corresponding data were recorded. These datasets were then classified using both the ICNN-LSTM model and the five benchmark models. All models achieved classification accuracy exceeding 64%. As illustrated in Figure 11, the results indicate a clear correlation between the helmet posture sensor data and the chinstrap tightness state. Moreover, deep learning models demonstrate superior feature extraction capabilities, enabling them to capture more discriminative patterns compared to shallow learning approaches.

(2)Fusion of Sensor Wear Data

In the application scenario involving fused sensor wear data, the data collected from the three sensors on each participant were concatenated into a unified dataset for classification training. This approach enabled the evaluation of fusion effects across different helmet wearing states. For all experimental tasks, classification performance comparisons were conducted using both the proposed ICNN-LSTM model and the five benchmark models. The results, as presented in Figure 12, indicate that the ICNN-LSTM model maintains excellent classification performance under the multi-sensor fusion setting.

(3)Analysis of Combined Multi-Sensor Ablation Experiments

To assess the effectiveness and contribution of the bionic strategy proposed in this study—based on a multi-sensor perception framework—an ablation study was conducted. Specifically, data collected at a robot operating speed of 0.05 m/s were used to evaluate model performance under various sensor configurations (single sensor, dual sensors, and full three-sensor fusion). The experimental outcomes were systematically documented in Table 6 (where “1” denotes data from Sensor 1 alone, “1 + 2” represents the fusion of Sensors 1 and 2, and so forth). The results show that the ICNN-LSTM model consistently achieves either the best or joint-best performance across nearly all sensor combinations. Moreover, the classification accuracy of other models also improves with the inclusion of additional sensors. These findings confirm that the proposed bionic strategy, together with the deep fusion architecture, effectively and robustly captures discriminative features from core sensors by analyzing vibration signals at varying frequencies. This functional mechanism emulates the role of spider leg bristle structures with differing lengths, thereby validating the efficacy and robustness of the bionic design.

### 5.3. Confusion Matrix

The classification results corresponding to a robot operating speed of 0.1 m/s (the most frequently utilized speed in practical scenarios) were selected for plotting the confusion matrix, enabling an evaluation of the diagnostic accuracy of various models across the three chinstrap tightness states. As presented in Figure 13, the *x*-axis and *y*-axis represent the predicted labels and true labels, respectively. Analysis of the confusion matrix and associated accuracy metrics reveals that RF, KNN, and BP exhibit relatively poor performance in identifying the “Not fastened properly” state, with KNN yielding the lowest accuracy. This can be attributed to the reduced rigidity in the connection between the head and the helmet lining when the chinstrap is loosely fastened, leading to significant interference in the helmet’s oscillation patterns and consequently complicating the extraction of reliable vibration features.

While RNN and LSTM demonstrate notable improvements in overall recognition accuracy, the classification performance for the “Not fastened properly” state remains suboptimal. In contrast, the proposed ICNN-LSTM model achieves superior classification performance across all three states. Both individual class accuracy and overall accuracy are significantly higher compared to those of the other models, highlighting the effectiveness of the improved architecture in capturing discriminative features under varying chinstrap conditions.

## 6. Conclusions

This study explores the identification of chinstrap tightness states in safety helmets using biomimetic-inspired methodologies. A practical data collection strategy and state classification model were developed. By analyzing the mechanics of safety helmet usage, a MEMS-based posture data acquisition approach inspired by spider leg bristles was proposed. Multiple MEMS posture sensors were mounted on the helmet’s upper surface to monitor dynamic posture and vibration characteristics. To extract relevant features, an ICNN-LSTM feature extraction and classification framework was introduced. This model combines adaptive CNN with LSTM networks to enhance chinstrap tightness state classification.

In single-sensor data classification, the ICNN-LSTM model achieved over 92% accuracy for Sensor 3 under all conditions, outperforming other fault diagnosis models in noise robustness. In multi-sensor data classification, the model reached 96% accuracy, demonstrating its strong generalization ability. The model is particularly effective in multi-sensor acquisitions inspired by spider leg bristles.

Despite strong performance in controlled environments, some challenges remain for practical implementation. The dependency on pre-collected labeled datasets may limit applicability in data-scarce scenarios, and future work should explore unsupervised or semi-supervised learning approaches. Environmental noise and variations in operational conditions can affect sensor accuracy, so efforts should focus on improving sensor design and adapting machine learning algorithms to dynamic conditions. Future research will prioritize integrating sensors seamlessly into helmet structures and conducting comprehensive field testing to validate the method, address operational issues, and guide improvements. This research aims to advance safety helmet monitoring systems for enhanced occupational safety.

## Figures and Tables

**Figure 1 biomimetics-10-00570-f001:**
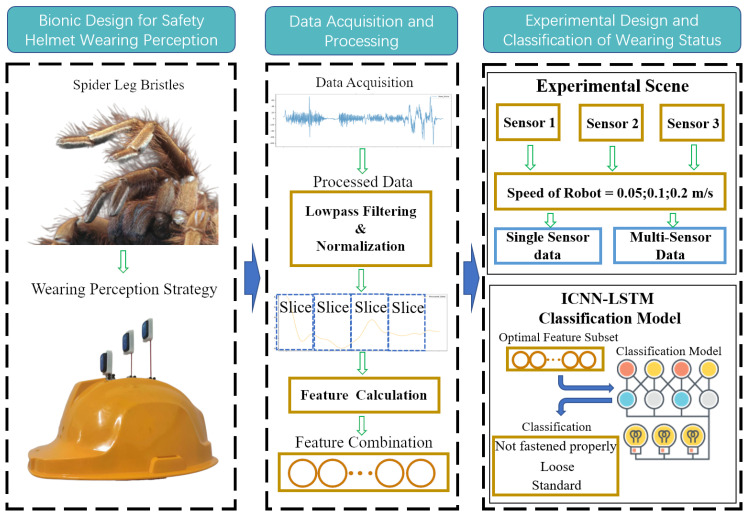
Overall framework of this research.

**Figure 2 biomimetics-10-00570-f002:**
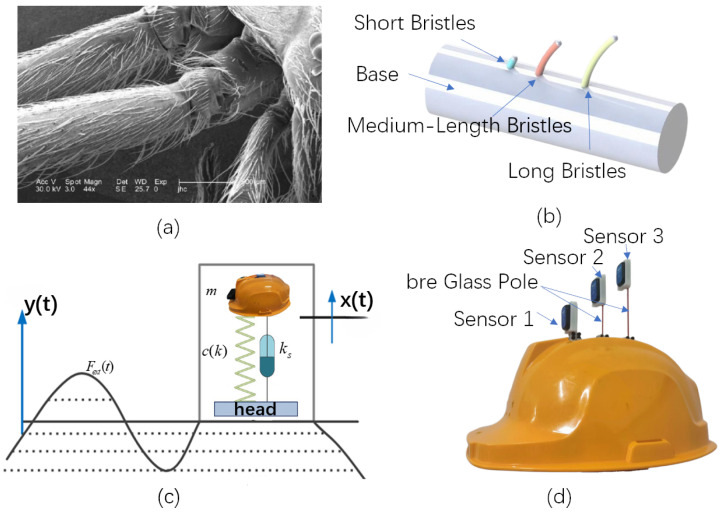
Bionic design and perception mechanism for helmet wearing detection; (**a**) micrograph of spider leg setae; (**b**) simplified model of leg setae; (**c**) equivalent damping model for helmet wearing; (**d**) bionic design of helmet setae.

**Figure 3 biomimetics-10-00570-f003:**
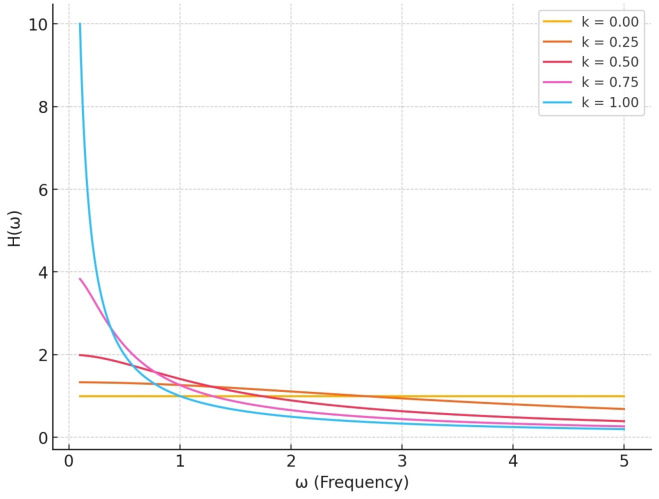
Transfer functions under different chinstrap tightness coefficients *k*.

**Figure 4 biomimetics-10-00570-f004:**
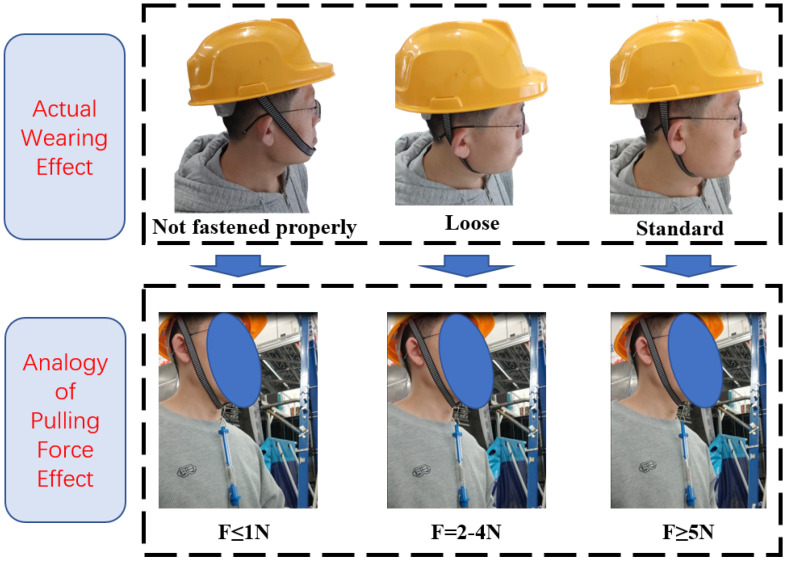
Three wearing states and their analogous effects.

**Figure 5 biomimetics-10-00570-f005:**
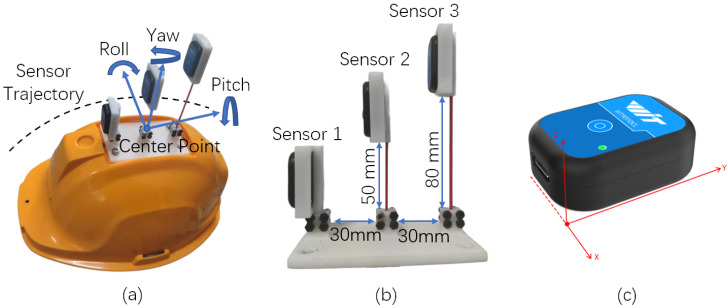
Wearable sensory helmet system. (**a**) MEMS attitude sensors layout; (**b**) installation position of sensors; (**c**) sensor coordinate system.

**Figure 6 biomimetics-10-00570-f006:**
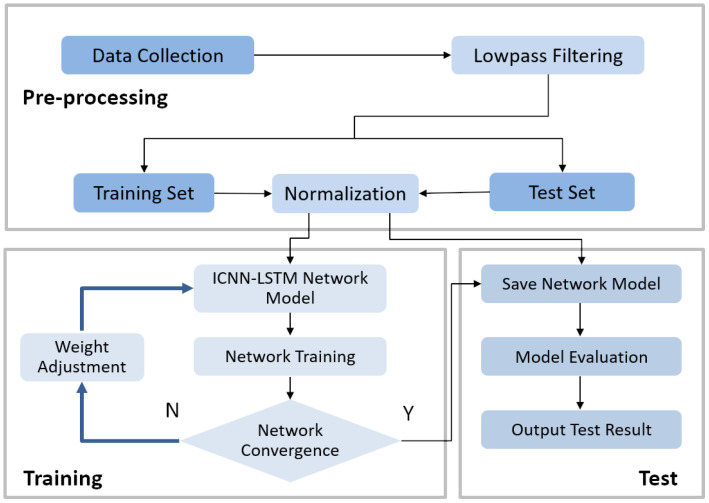
Flowchart of classification for chin strap tightness of helmets.

**Figure 7 biomimetics-10-00570-f007:**
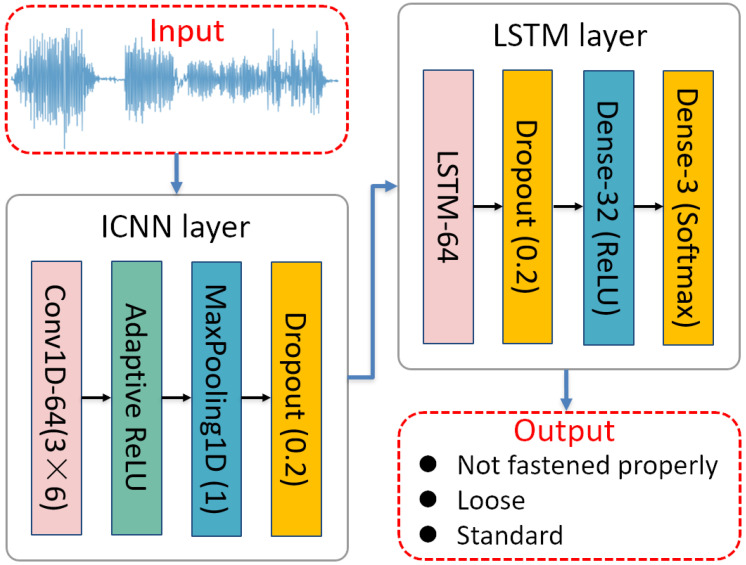
ICNN-LSTM network architecture.

**Figure 8 biomimetics-10-00570-f008:**
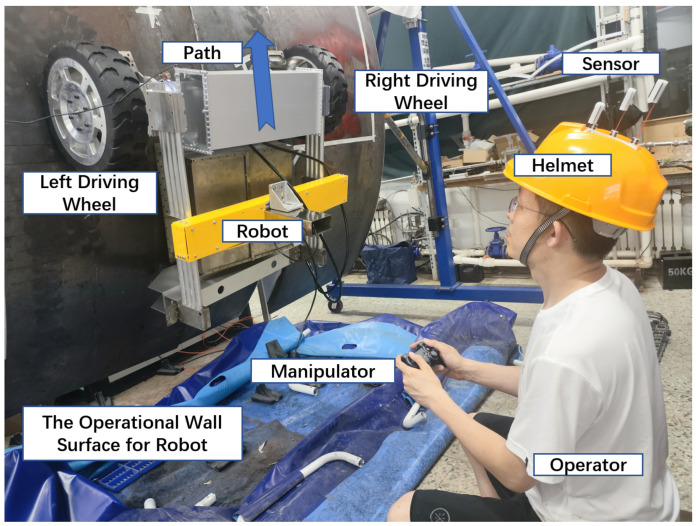
Wall-climbing robot control system.

**Figure 9 biomimetics-10-00570-f009:**
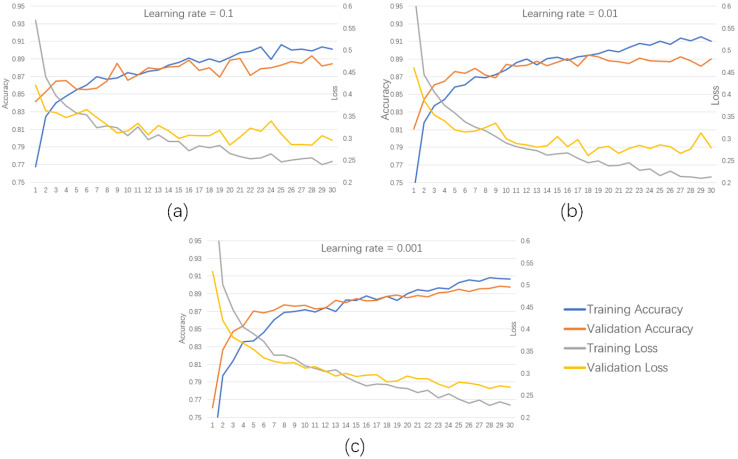
Comparison of accuracy and loss during training with different learning rates; (**a**) accuracy and loss rate with a learning rate of 0.1; (**b**) accuracy and loss rate with a learning rate of 0.01; (**c**) accuracy and loss rate with a learning rate of 0.001.

**Figure 10 biomimetics-10-00570-f010:**
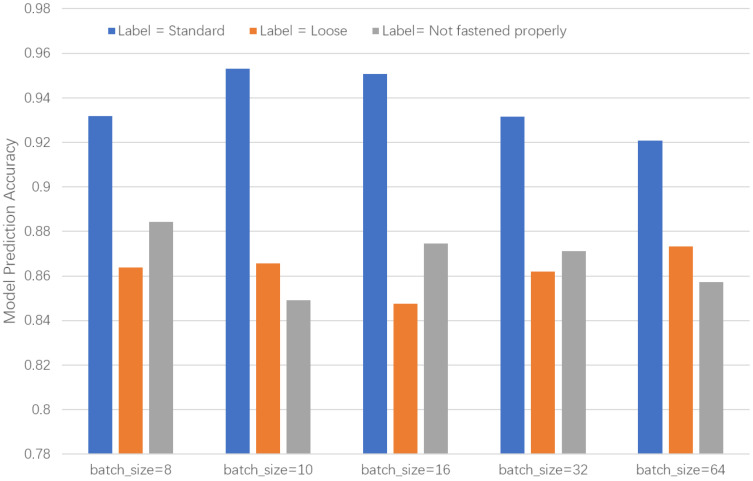
Accuracy rates corresponding to different training batch values for the three labels.

**Figure 11 biomimetics-10-00570-f011:**
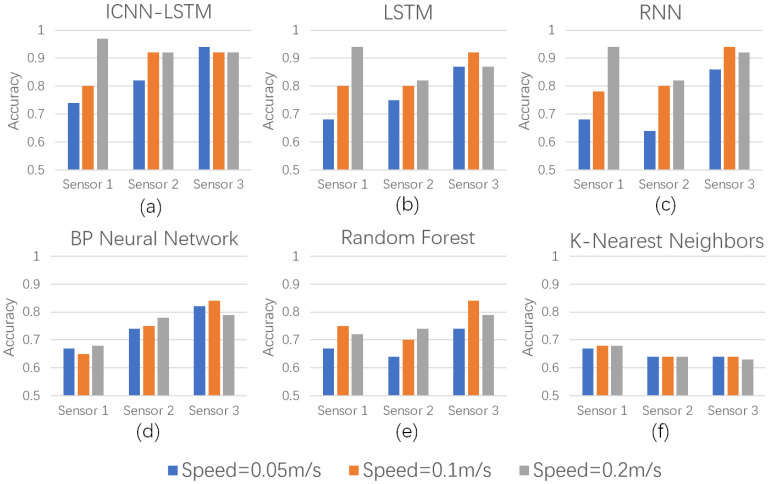
Comparison of classification accuracy for each model using single sensor at different speeds; (**a**) ICNN-LSTM model; (**b**) LSTM model; (**c**) RNN model; (**d**) BP model; (**e**) RF model; (**f**) KNN model.

**Figure 12 biomimetics-10-00570-f012:**
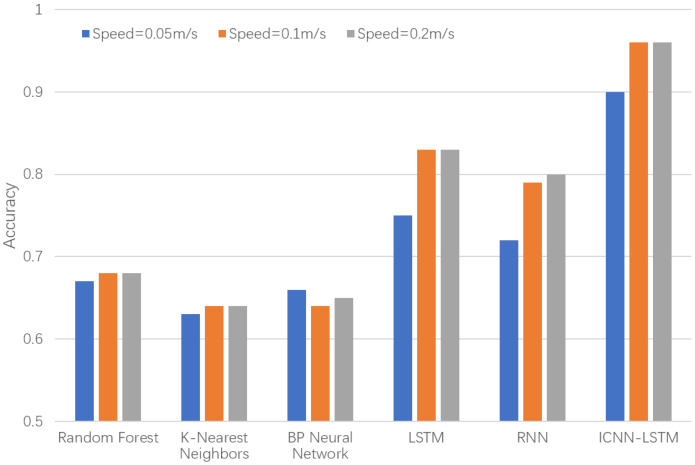
Comparison of classification accuracy for fused sensor data at different speeds.

**Figure 13 biomimetics-10-00570-f013:**
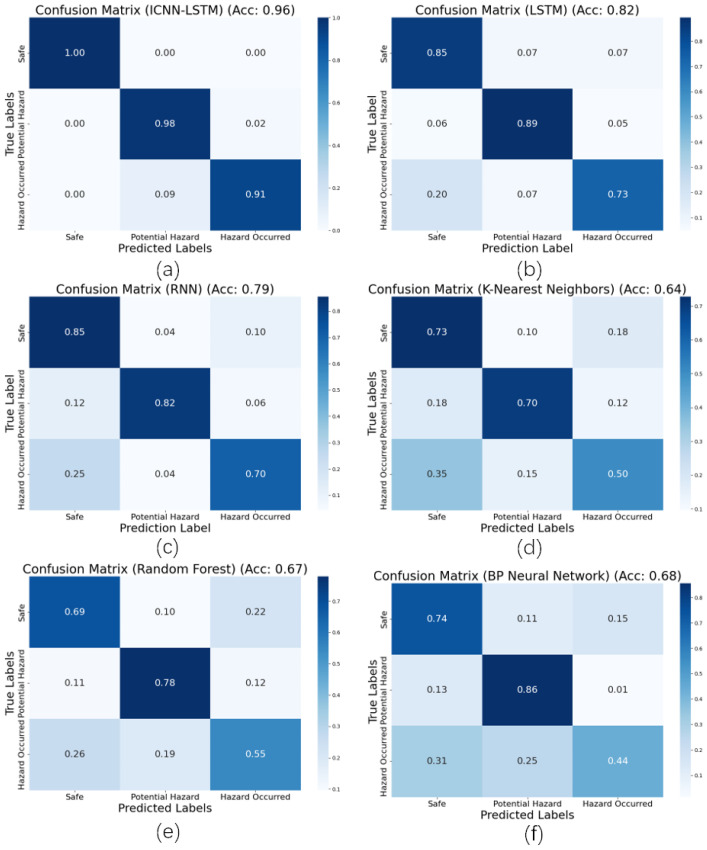
Comparison of confusion matrices for different models: (**a**) ICNN-LSTM model; (**b**) LSTM model; (**c**) RNN model; (**d**) KNN model; (**e**) RF model; (**f**) BP model.

**Table 1 biomimetics-10-00570-t001:** Sensor installation parameters.

Number	Installation Location	Fiber Rod Diameter (mm)	Fiber Rod Length (mm)
1	Front	N/A	N/A
2	Middle	2	50
3	Rear	2	80

**Table 2 biomimetics-10-00570-t002:** MEMS attitude sensor parameters and precision.

Serial Number	Parameters	Precision
1	Measurement Range	±2000%/s
2	Sampling Frequency	0.2–100 kHz
3	Resolution: 0.061	0.061 (%/s)/(LSB)
4	Static Zero Bias	±0.5∼1%/s
5	Temperature Drift	±0.005∼0.015(%/s)/°C
6	Sensitivity	≤0.015/s RMS

**Table 3 biomimetics-10-00570-t003:** Class names and categories for all feature dimensions.

Feature Dimension	Feature Type	Category
1	Mean	Time Domain
2	Standard Deviation	Time Domain
3	Maximum Value	Time Domain
4	Minimum Value	Time Domain
5	Norm	Time Domain
6	Energy	Time Domain
7	Kurtosis	Time Domain
8	Skewness	Time Domain
9	Simple Mean Absolute Value	Time Domain
10	Autocorrelation	Time Domain
11	Autocorrelation Lag 2	Time Domain
12	Autocorrelation Lag 3	Time Domain
13	Mean Power Frequency	Frequency Domain
14	Median Frequency	Frequency Domain
15	Total Power	Frequency Domain
16	Maximum Power Spectral Density	Frequency Domain
17	Zero Crossing Rate	Frequency Domain

**Table 4 biomimetics-10-00570-t004:** Personal information of participants.

Number	Age	Gender	Height (cm)	Weight (kg)	BMI	Body Fat Percentage (%)
1	24	Female	168	52	19.8	22
2	24	Male	175	75	24.4	18
3	25	Male	176	70	22.6	16
4	26	Male	180	78	24.1	19
5	24	Male	182	78	23.5	20
6	25	Male	177	82	26.2	22
7	24	Male	175	75	24.4	18
8	26	Male	180	75	23.1	17
9	27	Male	180	82	25.3	23
10	25	Male	178	78	24.6	21

**Table 5 biomimetics-10-00570-t005:** Main parameters of the ICNN-LSTM model.

Number	Hyperparameter	Value	Number	Hyperparameter	Value
1	Conv1D Filters	64	6	LSTM Units	64
2	Conv1D Kernel Size	3	7	Dropout Rate (LSTM)	0.2
3	Conv1D Activation	ReLU	8	Dense Layer Units	32
4	MaxPooling1D Pool Size	1	9	Output Activation	Softmax
5	Dropout Rate (Conv1D)	0.2	10	Shuffle	True

**Table 6 biomimetics-10-00570-t006:** Ablation experiment result record.

Model	1	2	3	1 + 2	1 + 3	2 + 3	1 + 2 + 3
ICNN-LSTM	0.8	0.82	0.94	0.78	0.84	0.87	0.94
LSTM	0.68	0.75	0.87	0.72	0.75	0.78	0.86
RNN	0.68	0.64	0.86	0.68	0.73	0.76	0.86
BP	0.67	0.74	0.82	0.70	0.74	0.78	0.81
RF	0.67	0.64	0.74	0.64	0.70	0.70	0.75
KNN	0.49	0.64	0.64	0.60	0.55	0.63	0.64

## Data Availability

The datasets generated during and or analyzed during the current study are not publicly available due to the data also forms part of an ongoing study.

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
