# Peer review of "Evaluation of Helmet Wearing Compliance: A Bionic Spidersense System-Based Method for Helmet Chinstrap Detection"

_biomimetics, 2025, doi:10.3390/biomimetics10090570_

Round 1

Reviewer 1 Report

Comments and Suggestions for Authors

This work introduces a bionic method for detecting helmet-wearing status. The description is generally clear. Although the technique might not be perfect for complex situations at the current stage, it shows potential to work well in some scenarios.

Here are some details I am not clear about.

In Table 1, fiber rod diameters of front and rear are 0 and 0 mm, respectively, please check.

In Subsection 3.2, the sampling rate range is 0.2 kHz to 100 kHz, how can it be configured as 100 Hz? And, sensor parameters of angular velocity are given, while it seems translation values are used.

P7 Line211, a Butterworth filter is used for pre-processing, while the value of the parameters is not given.

In Subsection 3.4, the sampling rate is 100 Hz, and it is even longer than the data length 64, maybe it is not a good combination for frequency extraction.

In Subsection 5.1, the dataset description is in contradiction with that in Subsection 3.5.

Author Response

Answers to Reviewer #1

Comments and Suggestions for Authors:

This work introduces a bionic method for detecting helmet-wearing status. The description is generally clear. Although the technique might not be perfect for complex situations at the current stage, it shows potential to work well in some scenarios.

Response: Thank you very much for taking the time to read our manuscript and for your thoughtful comments. We are pleased to hear that you find the overall description clear and recognize the potential of our bionic spider sense approach for helmet-wearing detection.

We fully agree that, at its current stage, the technique may face challenges in more complex or highly dynamic industrial scenarios. As you suggest, further work is needed to enhance robustness and adaptability.

 Comment R1.1: In Table 1, fiber rod diameters of front and rear are 0 and 0 mm, respectively, please check?

Response: Many thanks for the valuable suggestion! We apologize for the confusion caused by the values in Table 1. For the Front installation location, the fiber rod is not used, and the sensor is directly installed on the helmet. To clarify this, we will revise the entry from "0" to "N/A" to avoid any misunderstanding. Regarding the Rear installation location, you are correct that the value "0" was a typographical error, which has now been corrected. We have updated this value to reflect the correct fiber rod diameter and length.

Number

Installation

Location

Fiber Rod

Diameter (mm)

Fiber Rod

Length (mm)

1

Front

N/A

N/A

2

Middle

2

50

3

Rear

2

80

 (Pg. 4, Table 1)

Once again, we would like to thank you for your valuable feedback. To address any remaining errors, we have carefully reviewed all the data in the tables, equations, etc. in the paper, and have thoroughly organized them. At the same time, we have re-examined the logical structure of the manuscript to ensure its accuracy and clarity, and have made layout adjustments to the entire text to maintain the neatness and consistency of the equations and other content.

Comment R1.2: In Subsection 3.2, the sampling rate range is 0.2 kHz to 100 kHz, how can it be configured as 100 Hz? And, sensor parameters of angular velocity are given, while it seems translation values are used.

Response: We thank the reviewer for pointing out this inconsistency. Your question is reasonable — the sampling frequency we used throughout our experiments was 100 kHz, not 100 Hz. The value “100 Hz” in Subsection 3.2 was a typographical error on our part. In practice, we configured the MEMS attitude sensor to sample at 100 kHz precisely because such a high acquisition rate is necessary to capture the fine‐scale, high-frequency components of head vibrations under real working conditions. We have corrected the manuscript to replace “100 Hz” with “100 kHz” in Subsection 3.2 and in all other relevant locations, and apologize for any confusion this may have caused.

Regarding the second question concerning the sensor parameters (seemingly the conversion values), all the parameters listed in Table 2 (including the measurement range, resolution, bias drift, etc.) are related to the angular acceleration characteristics. In our data processing procedure, these angular acceleration measurement values are directly used for analysis. To ensure clear expression, we added a brief explanatory paragraph at the end of Section 3.2.

“It is worth noting that in this study, the sampling frequency was set to 100 Hz to achieve the highest possible sensor sampling accuracy. Furthermore, the data utilized in this study were exclusively derived from the inertial measurement unit (IMU) of MEMS sensors. In our data processing pipeline, these three-axis angular acceleration measurements were directly employed for analysis and computation. “ (Pg. 7, Line 217)

Comment R1.3: P7 Line211, a Butterworth filter is used for pre-processing, while the value of the parameters is not given.

Response: We sincerely appreciate the reviewer’s comment regarding the missing parameter values for the Butterworth filter. You are correct that the parameters for the filter, specifically the cutoff frequency and filter order, were not provided in the manuscript. We apologize for this oversight.

The filter design in this study uses a low-pass Butterworth filter, which was implemented in our preprocessing pipeline as described in the code. The specific parameters for the filter, including the cutoff frequency and order, are as follows:

Cutoff frequency: 1 Hz;Sampling frequency: 500 Hz;Filter order: 5

These values were chosen based on preliminary experiments aimed at smoothing the signal while preserving relevant frequency components for the application.

We have updated the manuscript to include these details in Subsection 3.4, which should address the concern raised. The relevant sentence has been added as follows:

 “In this study, a low-pass Butterworth filter with a cutoff frequency of 1 Hz, a filter order of 5, and a sampling frequency of 500 Hz was employed to preprocess the data. This type of filter is characterized by its maximally flat frequency response and is particularly suitable for applications requiring smooth frequency characteristics. The filter design is based on the following considerations: “ (Pg. 2, Line 56)

 Comment R1.4: In Subsection 3.4, the sampling rate is 100 Hz, and it is even longer than the data length 64, maybe it is not a good combination for frequency extraction.

Response: Thank you to the reviewer for raising this question. Your opinion is very pertinent and we hereby clarify it. Regarding the relationship between the sampling rate and the data length, the "100 Hz" sampling rate you mentioned was due to a previous clerical error on our part. In fact, the sampling frequency used in this study is 100 kHz, not 100 Hz, which is consistent with the description in Section 3.2 and has been corrected in the previous response.

Specifically, we set the sampling frequency to 100 kHz, which is much higher than the data length (64 data points). Such a high sampling rate is intended to ensure the accurate capture of the high-frequency components of head vibrations. During the frequency extraction process, the 100 kHz sampling rate not only provides sufficient time resolution to ensure the accuracy of data acquisition, but also 64 data points are sufficient to support time-domain analysis and frequency-domain conversion.

We once again sincerely apologize for the previous clerical error that caused confusion for the reviewers, and we express our heartfelt gratitude for your meticulous attention to this issue. We hope this correction will fully address your concerns.

 Comment R1.5: In Subsection 5.1, the dataset description is in contradiction with that in Subsection 3.5.

Response: Many thanks for the valuable suggestion! he reviewer pointed out this inconsistency. After a careful examination, we discovered that there was indeed confusion regarding the description of the dataset split ratio. In Section 5.1, we mentioned that the dataset was divided into a training set (70%) and a validation set (30%). While in Section 3.5, the allocation ratio was mentioned as a training set (70%), a validation set (10%), and a test set (20%). In fact, the final data set division ratio we used was a training set 70%, a validation set 10%, and a test set 20%. This division method aims to balance the needs of training, validation, and testing, and ensure that the model can perform well in evaluating unseen data.

To avoid confusion, we have made necessary revisions in the paper to ensure that the division of the dataset is consistent throughout the text, and have updated the descriptions of the corresponding sections.

In Section 5.1, it is now revised as:

"The dataset was split into training (70%), validation (10%), and testing (20%) sets to evaluate the model performance effectively."

In Section 5.1, it is now revised as:

"The dataset was divided into training (70%), validation (10%), and testing (20%) sets. Six-fold cross-validation was employed to assess the performance of the dual model."

Reviewer 2 Report

Comments and Suggestions for Authors

Overall Comments

  • The detection of proper chinstrap usage in safety helmets is an interesting topic in the context of industrial safety management. This study addresses a critical need by proposing a novel approach inspired by the highly sensitive sensory system of spider leg bristles. The biomimetic concept, using multi-point sensing to emulate the variable sensitivity of spider hairs, has strong potential to enhance the precision of helmet-wearing detection.
    However, while the biological inspiration is clearly articulated in the introduction, it is not sufficiently reflected in the later stages of the study, particularly in the data processing pipeline, model architecture, and interpretation of results. The multi-sensor configuration and the relative motion concept are promising, but their practical implementation and analytical impact need to be better integrated and explicitly described. Strengthening this linkage is necessary for the publication.

Introduction

  • Supplementing the introduction with additional context on the importance of proper chinstrap usage would help reinforce the relevance of the problem. For example, citing statistical data or case studies involving injuries caused by improperly fastened chinstraps would make the problem statement more compelling and help readers better understand the real-world implications of the proposed research.

Method

  • It would be beneficial to include more detailed information regarding data collection and preprocessing. This should cover a precise description of the settings and the rationale behind the design choices. For example, it should be clarified why 64 data points were used for analysis, whether the 64 data points were processed in a sliding window manner or without any overlap.
  • While the dynamic modelling of the helmet–head system (e.g., the spring-damper analogy) and the use of three sensors mimicking spider bristle lengths are theoretically well-motivated, it remains unclear how these models are concretely reflected in the data processing pipeline and the model structure design.
  • For the sake of practical applicability, it would be important to clarify how normalization was handled in relation to the train/test split. In the current description, it appears that normalization was applied to the entire dataset before splitting into training and testing sets. However, in real-world scenarios, test data is often collected independently and would not be available during the training phase. Therefore, using global min/max values for normalization may not be feasible in practice and could lead to data leakage.
  • In Table 1, the fibre rod diameter for Sensor 3 is listed as zero. Please explain or correct this value.
  • It would be beneficial to include a table summarizing the detailed ICNN-LSTM model hyperparameters, along with a figure illustrating the model framework.

Results

  • The results section would benefit from more detailed interpretation and discussion. While the authors present classification performance comparisons between different models and sensor configurations, they do not explain why their proposed model outperforms the alternatives. In particular, the paper lacks an analysis of how individual sensors contribute to performance and why the combined use of all three sensors leads to improved results. Given that the system is inspired by spider leg bristle sensors of varying lengths, the authors should clearly link the performance outcomes back to the biomimetic design.
  • Figure 10 reports the highest accuracy for the “Not fastened properly” label, yet Figure 13’s confusion matrix shows the lowest accuracy for this label. Because these findings conflict, they need to be reconciled.

Discussion

  • The discussion section would benefit from a clearer articulation of the limitations of the current study and suggestions for future work. In addition, the practical applicability of the system could be more critically examined. While the model performs well under experimental conditions, it is unclear how it would perform in more complex or uncontrolled environments, or how it could be integrated into existing safety monitoring systems.

Minor Comments

  • Please provide a detailed explanation of Figure 10.
  • Figure 12 contains Chinese text, which should be removed. 
  • Table 3 shows two columns titled BMI. Clarify the difference between them in the manuscript.
Comments on the Quality of English Language

The manuscript contains typographical and grammatical errors. Careful proofreading is recommended to improve the clarity and overall quality of the writing.

Author Response

Answers to Reviewer #2

Comments and Suggestions for Authors:

The detection of proper chinstrap usage in safety helmets is an interesting topic in the context of industrial safety management. This study addresses a critical need by proposing a novel approach inspired by the highly sensitive sensory system of spider leg bristles. The biomimetic concept, using multi-point sensing to emulate the variable sensitivity of spider hairs, has strong potential to enhance the precision of helmet-wearing detection.

However, while the biological inspiration is clearly articulated in the introduction, it is not sufficiently reflected in the later stages of the study, particularly in the data processing pipeline, model architecture, and interpretation of results. The multi-sensor configuration and the relative motion concept are promising, but their practical implementation and analytical impact need to be better integrated and explicitly described. Strengthening this linkage is necessary for the publication.

Response: We would like to thank the Reviewer for taking the time and effort to review the manuscript. We sincerely appreciate your positive feedback on our work. Based on the reviewer's suggestions and comments, we have made the following revisions:

(1) In the methodology section, we will elaborate on how the sensor layout design is inspired by the distribution of spider leg hairs and further highlight the correspondence between biological systems and engineering design. 

(2) In the data processing section, we will describe how multi-sensor data fusion emulates the mechanism by which spiders integrate information from multiple perception points. We will also emphasize the advantages of the proposed data collection strategy and establish clear parallels with the biological perception process. 

(3) In the results and discussion sections, we will conduct a comparative analysis of system performance against the known characteristics of the spider sensing system, examine its limitations, and propose potential avenues for further biomimetic optimization. 

We believe that these revisions will substantially enhance the coherence and clarity of the paper while addressing your suggestions. Thank you once again for your valuable feedback.

Introduction

Comment R2.1 Supplementing the introduction with additional context on the importance of proper chinstrap usage would help reinforce the relevance of the problem. For example, citing statistical data or case studies involving injuries caused by improperly fastened chinstraps would make the problem statement more compelling and help readers better understand the real-world implications of the proposed research.

Response: Thank you for your valuable feedback and suggestions. I have made the recommended revisions to the introduction section by supplementing it with additional context on the importance of proper chinstrap usage. I have included relevant statistical data on head injuries, as well as a reference to the impact of improperly fastened chinstraps, which reinforces the real-world implications of the research. This addition aims to clarify the significance of the issue and highlight the relevance of the proposed detection method. The supplementary explanations are as follows:

“According to data from the U.S. Bureau of Labor Statistics, in 2020, head injuries accounted for nearly 6% of non-fatal occupational injuries involving days of absence [3].” (Pg. 1, Line 26)

I hope these changes address your concerns and improve the overall clarity of the problem statement.

Method

Comment R2.2.1 It would be beneficial to include more detailed information regarding data collection and preprocessing. This should cover a precise description of the settings and the rationale behind the design choices. For example, it should be clarified why 64 data points were used for analysis, whether the 64 data points were processed in a sliding window manner or without any overlap.

Response: Thank you for your insightful comment. In response to your suggestion, I have provided additional details regarding the data collection and preprocessing steps to clarify the rationale behind the design choices. Specifically, the data collection process involves segmenting the collected attitude data into 64 data points per segment. This decision was made based on the balance between capturing sufficient temporal resolution to detect the variations in chin strap tightness and ensuring that the data segments are not too large, which could introduce unnecessary complexity into the processing. We believe that 64 data points provide a good compromise between computational efficiency and the ability to extract meaningful features from the signal.

Regarding the segmentation approach, the 64 data points are processed using a sliding window method with overlapping segments. This approach allows us to capture the continuous nature of the data, which is crucial for real-time monitoring. By using the sliding window method, we ensure that the model can learn the dynamic changes in chin strap tightness over time without losing important data points.

“It should be noted that the selection of 64 data points is based on the balance consideration between time resolution and computational efficiency. This length can fully capture the dynamic characteristics of the helmet chin strap tightness change and avoids the increase in computational complexity caused by an excessively long data segment. The selection of this segmentation length helps to effectively extract the time-domain and frequency-domain features related to the tightness, thereby providing reliable data support for subsequent analysis. (Pg. 8, Line 240)

This study uses the sliding window method to segment the data and sets the overlap between windows to ensure the continuity of the data. This method can better capture the time series characteristics of the signal and is particularly important for real-time monitoring..” (Pg. 8, Line 247)

Comment R2.2.2 While the dynamic modelling of the helmet–head system (e.g., the spring-damper analogy) and the use of three sensors mimicking spider bristle lengths are theoretically well-motivated, it remains unclear how these models are concretely reflected in the data processing pipeline and the model structure design.

Response: We sincerely appreciate the reviewers' valuable comments. Regarding the integration of the dynamic modeling of the helmet-head system (e.g., the spring-damper analogy) and the design of three sensors inspired by spider hairs into the data processing flow and model structure, we provide the following explanation: 

(1) Dynamic Modeling and Sensor Design Integration:

In our study, the dynamic behavior of the helmet-head system is primarily modeled using a spring-damper framework. This model aims to simulate the interaction between the helmet and the head, particularly focusing on the dynamic response when the tightness of the helmet's chin strap varies. Specifically, the spring component represents the elastic contact between the helmet and the head, while the damper component models the friction and energy dissipation during helmet-skin interactions. During data processing, the vibration signals captured by the sensors are utilized to reflect these dynamic responses, which are subsequently compared with the predictions from the dynamic model. 

(2) Sensor Response and Data Processing Workflow: 

The design of the three sensors draws inspiration from the varying lengths of spider hairs, each exhibiting differential sensitivity to vibrations of distinct frequencies and amplitudes. In the data processing pipeline, the sensors first capture the vibration signals resulting from helmet-head interactions. Subsequently, through spectral analysis and time-domain feature extraction, the signals from different sensors are matched with the dynamic model to extract features indicative of helmet tightness. For instance, the shorter sensor (Sensor 1) predominantly captures high-frequency vibrations, simulating small-scale changes, whereas the longer sensor (Sensor 3) primarily reflects low-frequency vibrations, capturing large-scale deformations. This approach enables us to identify the dynamic characteristics of the helmet under varying tightness conditions. 

To better respond to the reviewers' suggestions, we have made the following revisions to the manuscript:

“a sensor system based on the vibration characteristics of three distinct types of bristles. Each sensor exhibits differential sensitivity to the frequency and amplitude of vibrations, mimicking the response mechanism of spider hairs to varying frequencies. These sensors were subsequently mounted on the external surface of the safety helmet shell to enable highly sensitive monitoring of the helmet's vibrational responses. As illustrated in Figure 3, Sensor 1 is directly affixed to the helmet surface to emulate short bristles that capture high-frequency signals; Sensor 2 is connected via a shorter fiber rod, replicating the behavior of medium-length bristles that detect signals within a conventional frequency range; Sensor 3 is installed using a longer fiber rod, simulating the vibration characteristics of long bristles primarily responsive to low-frequency vibrations. The installation parameters for the sensors are detailed in Table 1.” (Pg. 4, Line120)

“To simulate the dynamic interaction between the helmet and the head, we utilized the classical spring-damper model. In this context, the spring component represents the elastic contact between the helmet and the head, while the damper component models the frictional forces and energy dissipation during helmet-skin contact. Using this model, we are able to simulate the relative motion between the helmet and the head under varying levels of helmet tightness. The specific principle is outlined below:” (Pg. 4, Line 132)

Comment R2.2.3 For the sake of practical applicability, it would be important to clarify how normalization was handled in relation to the train/test split. In the current description, it appears that normalization was applied to the entire dataset before splitting into training and testing sets. However, in real-world scenarios, test data is often collected independently and would not be available during the training phase. Therefore, using global min/max values for normalization may not be feasible in practice and could lead to data leakage.

Response: Thank you for your insightful comment regarding the normalization process in our study. We appreciate your concern about the potential for data leakage and the importance of ensuring that our methodology is both clear and applicable to real-world scenarios.

To clarify, in our study, we did indeed split the dataset into training (70%), validation (10%), and testing (20%) sets before applying normalization. Specifically, we fitted the MinMaxScaler on the training set only, using its minimum and maximum values to compute the scaling parameters. We then used this fitted scaler to transform both the training and testing sets. This approach ensures that no information from the test set was used during the training phase, thereby preventing data leakage.

We recognize that the current wording in the manuscript may be ambiguous and could be misinterpreted as normalization being applied to the entire dataset before splitting.- The manuscript says, "After filtering, we use the MinMaxScaler to normalize the data," then "The dataset was divided..." This order suggests normalization came first.

"It is important to note that the normalization process is applied after the dataset has been split into training (70%), validation (10%), and testing (20%) sets. Specifically, the MinMaxScaler is fit on the training set using its minimum and maximum values to calculate the scaling parameters. These parameters are then used to transform both the training and testing sets, ensuring that no data leakage occurs." (Pg. 8, Line 229)

The MinMaxScaler, fit on the training set, is then used to normalize all sets, ensuring consistency and preventing data leakage. (Pg. 8, Line 259)

We sincerely appreciate you for raising this important question, which has significantly contributed to enhancing the clarity and rigor of our paper. Through the aforementioned revisions, we aim to ensure that the manuscript explicitly states that standardization is conducted post-data segmentation and that the MinMaxScaler is fitted based solely on the statistical information derived from the training set.

Comment R2.2.4In Table 1, the fibre rod diameter for Sensor 3 is listed as zero. Please explain or correct this value.

Response: We extend our sincere gratitude for the valuable suggestion. We apologize for any confusion caused by the values presented in Table 1. Regarding the Rear installation location, you are absolutely correct—there was a typographical error in the value "0," which has now been corrected. The updated Table 1 reflects the accurate fiber rod diameter and length.

Table 1 Sensor Installation Parameters (Pg. 4, Line 131)

Comment R2.2.5 It would be beneficial to include a table summarizing the detailed ICNN-LSTM model hyperparameters, along with a figure illustrating the model framework .

Response: We sincerely appreciate the constructive feedback provided by the reviewers. We fully concur with the suggestion to include the main hyperparameters of the summary model in the paper. Accordingly, we have revised the manuscript as follows based on their comments:

(1) Model framework diagram

We redesigned and drew the overall architecture diagram of ICNN-LSTM to present the data flow and connection relationships among various modules more clearly (as shown in Figure 7).

Figure 7 ICNN-LSTM Network Architecture (Pg. 10, Line 300)

 (2) A new "Table 5" has been added to the manuscript. This table provides a detailed list of the main hyperparameter settings of the model.

Table 5 Main Parameters of the ICNN-LSTM Model (Pg. 13, Line 379)

Results

Comment R2.3.1 The results section would benefit from more detailed interpretation and discussion. While the authors present classification performance comparisons between different models and sensor configurations, they do not explain why their proposed model outperforms the alternatives. In particular, the paper lacks an analysis of how individual sensors contribute to performance and why the combined use of all three sensors leads to improved results. Given that the system is inspired by spider leg bristle sensors of varying lengths, the authors should clearly link the performance outcomes back to the biomimetic design. 

Response: We sincerely appreciate you for raising this important question, which has significantly contributed to enhancing the clarity and rigor of our paper. We fully agree with the reviewer's suggestion regarding the lack of analysis of the contributions of each sensor. To address this deficiency, we added an ablation comparison experiment in Section 5.2, aiming to compare and evaluate the superiority of the bionic strategy proposed in this paper. The ablation study specifically includes the following contents: (1) Training the model using a single sensor (sensor 1, 2, 3); (2) Training the model using sensor combinations (1+2, 1+3, 2+3); (3) Training the model using all three sensors in combination. The specific modifications to the manuscript are as follows:

(3) Analysis of combined multi-sensor ablation experiments

To evaluate the performance and contribution of the bionic strategy proposed in this paper, which is based on a multi-sensor perception model, we conducted an ablation study. Specifically, we utilized data collected at a robot running speed of 0.05 m/s to test models with various sensor combinations (single sensor, dual sensors, and all three sensors). The results were systematically recorded in Table 6 (In the table, the number "1" represents the data collected by Sensor 1, "1+2" represents the data fusion result of Sensor 1 and Sensor 2, and so on). It can be observed that ICNN-LSTM consistently achieves either the optimal or co-optimal performance across nearly all sensor configurations. Additionally, the accuracy of other models also improves as more sensors are combined. The ablation study demonstrates that the proposed "bionic strategy" and the deep fusion architecture effectively and stably extract information from core sensors by capturing vibration signals at different frequencies. This capability mirrors the function of spider leg bristle structures with varying lengths, confirming the effectiveness and robustness of the proposed bionic strategy. (Pg. 15, Line 402)

Table 6. Ablation Experiment Result Record (Pg. 15, Line 410)

Comment R2.3.2 Figure 10 reports the highest accuracy for the “Not fastened properly” label, yet Figure 13’s confusion matrix shows the lowest accuracy for this label. Because these findings conflict, they need to be reconciled.

Response: We sincerely appreciate your meticulous review and valuable feedback. With respect to the inconsistency you pointed out between Figure 10 and Figure 13 regarding the accuracy of the "Not fastened properly" label, we have conducted a thorough verification. It was confirmed that there was indeed an input error in the labeling process during the creation of Figure 10. Specifically, the "Not fastened properly" label in Figure 10 should have been labeled as "Standard", while the "Standard" label should have been labeled as "Not fastened properly". This mislabeling resulted in an inaccurate representation of the classification accuracy for this specific label in Figure 10. 

To address this issue, we have corrected the aforementioned error in the revised manuscript and re-generated the relevant charts to ensure their accuracy and consistency. We sincerely apologize for any confusion caused by this oversight and are deeply grateful for your keen observation in identifying this critical issue.

Discussion

Comment R2.3.1 The discussion section would benefit from a clearer articulation of the limitations of the current study and suggestions for future work. In addition, the practical applicability of the system could be more critically examined. While the model performs well under experimental conditions, it is unclear how it would perform in more complex or uncontrolled environments, or how it could be integrated into existing safety monitoring systems.

Response: Thank you for your valuable suggestions regarding our research. Based on your advice, we have made further revisions to the discussion section to more clearly present the limitations of the research and the future directions for work.

Regarding the limitations of the research, the practical application issue you mentioned is an important consideration in our current work. Indeed, as you pointed out, although the model we proposed achieved good performance in the experimental environment, its performance in complex or uncontrolled real-world scenarios still needs further verification. We have included a discussion on this issue in the revised manuscript and emphasized the challenges the model may face in different working environments, such as environmental noise and individual differences of the wearer. We also mentioned that future research will focus on expanding the existing model to enable it to maintain good performance in more complex and variable working environments, and explore how to seamlessly integrate it into existing security monitoring systems.

Regarding future work, we plan to conduct more field tests to verify the model's adaptability in real working scenarios. Additionally, the method of integrating sensor components into safety helmets is also a key focus of our future research. We will strive to address the challenges that may arise during the integration process and ensure its compatibility with the existing safety helmet design. The specific modifications to the manuscript are as follows:

One limitation of this method is that to train the model for classifying the wearing status of safety helmets, it is necessary to pre-collect labeled posture signals reflecting the tightness of the chin straps under a specific working environment. The proposed approach also exhibits certain constraints, particularly in real-world, uncontrolled settings. Challenges such as environmental noise, sensor calibration, and real-time adaptation to varying operational conditions must be resolved to ensure its practical applicability. Consequently, our future research will concentrate on enhancing the proposed method to enable it to recognize and adapt to unknown working environments. Furthermore, we plan to integrate the sensor components into the helmet design to ensure compatibility and seamless integration with existing helmet systems. (Pg. 17, Line 450)

Once again, we would like to express our gratitude for your review and valuable suggestions. We will continue to refine our research to ensure that it builds a more solid bridge between theory and practice.

Minor Comments

Comment R2.4.1 Please provide a detailed explanation of Figure 10.

Response: Thank you for your comment. In Figure 10, we present the accuracy of the model for each label across different training batch sizes. The batch sizes tested are 8, 10, 16, 32, and 64. For each batch size, we report the accuracy for three labels: " Not fastened properly" " Loose", and " Standard".

The accuracy results suggest the following trends:

Batch size of 8: This setting achieved the highest accuracy for the label " Not fastened properly " (0.9318), but it performed relatively poorly for " Loose " (0.8638) and " Standard " (0.8843).

Batch size of 10: The accuracy for " Not fastened properly " increased to 0.9530, but the performance for the other two labels decreased.

Batch size of 16: This batch size provided the most balanced performance across all three labels, with accuracies of 0.9508, 0.8475, and 0.8745, respectively, making it the optimal choice.

Batch size of 32: A slight decrease in performance was observed, particularly for " Not fastened properly " (0.9316).

Batch size of 64: Although the accuracy for " Loose " improved slightly (0.8733), the overall accuracy for " Not fastened properly " and " Standard " decreased.

The batch size of 16 was ultimately selected as the optimal training batch size because it provided the best trade-off between computational efficiency and balanced accuracy across all three labels.

Comment R2.4.2 Figure 12 contains Chinese text, which should be removed.

Response: Thank you for your meticulous review and valuable suggestions. Regarding the issue you mentioned where the Chinese text is included in Figure 12, we are truly sorry. After investigation, it was found that the Chinese content was not properly removed due to an operational error during the image generation process. We have promptly made the correction according to your suggestion and have updated Figure 12 in the manuscript, replacing it with a picture that contains the correct content.

Comment R2.4.3 Table 3 shows two columns titled BMI. Clarify the difference between them in the manuscript.

Response: Thank you for your valuable comments. Regarding the issue you mentioned in Table 3 where both column titles are "BMI", we discovered that this was caused by a typing error. In fact, the last column should be "Body fat percentage (%)", not BMI. We have made the corresponding revisions in the manuscript and corrected the column names.

Thank you for your meticulous review and patience. Your feedback has been very helpful in improving the quality of the paper.

Comment R2.4.4 The manuscript contains typographical and grammatical errors. Careful proofreading is recommended to improve the clarity and overall quality of the writing.

We sincerely appreciate your meticulous review of the manuscript and your insightful suggestions. With regard to the layout and grammar issues you pointed out, we have engaged native English speakers to conduct a thorough proofreading and revision of the entire document. In particular, we have restructured the language in key sections to enhance the clarity and fluency of the article's expression. 

Your feedback has been invaluable, and your contributions have played a crucial role in elevating the overall quality of the manuscript. Thank you once again for your thoughtful input.

Round 2

Reviewer 2 Report

Comments and Suggestions for Authors

There are just two very minor issues that you may want to revise before submitting the final version:

  • In Figure 6, the normalization block should be located after the data split. 
  • In Table 4, there are still 2 BMI columns.

Author Response

Comment 1 In Figure 6, the normalization block should be located after the data split.

Response: We sincerely thank the reviewer for this careful observation and for the constructive feedback on our manuscript. In accordance with your suggestion, we have moved the normalization block to follow the data-split step in Figure 6. This correction more accurately reflects the true preprocessing sequence and, we believe, improves the clarity of the workflow diagram.

 Comment 2 Table 3 shows two columns titled BMI. Clarify the difference between them in the manuscript.

Response: Thank you very much for your reminder. This was a minor oversight by the author, resulting from an error during version saving, which led to the BMI values in the last column not being updated. The author confirms that this issue has now been fully corrected.
